# Influence of Femtosecond Laser Surface Modification on Tensile Properties of Titanium Alloy

**DOI:** 10.3390/mi15010152

**Published:** 2024-01-19

**Authors:** Kai Zhou, Xiaoyuan Nie, Xingbang Che, Han Xiao, Xuwen Wang, Junming Liao, Xu Wu, Can Yang, Chunbo Li

**Affiliations:** 1Sino-German College of Intelligent Manufacturing, Shenzhen Technology University, Shenzhen 518118, China; 2110412015@stumail.sztu.edu.cn (K.Z.); 201901010533@stumail.sztu.edu.cn (X.N.); 2210415005@stumail.sztu.edu.cn (H.X.); 2200411006@stumail.sztu.edu.cn (X.W.); 2310412022@stumail.sztu.edu.cn (J.L.); wuxu@sztu.edu.cn (X.W.); 2Key Laboratory of Advanced Optical Precision Manufacturing Technology of Guangdong Higher Education Institutes, Shenzhen Technology University, Shenzhen 518118, China; 2210412002@stumail.sztu.edu.cn

**Keywords:** TC4, femtosecond laser, surface modification, process parameters, mechanical properties

## Abstract

Titanium alloy components often experience damage from impact loads during usage, which makes improving the mechanical properties of TC4 titanium alloys crucial. This paper investigates the influence of laser scanning irradiation on the tensile properties of thin titanium alloy sheets. Results indicate that the tensile strength of thin titanium alloy sheets exhibits a trend of initial increase followed by a decrease. Different levels of enhancement are observed in the elongation at break of a cross-section. Optimal improvement in the elongation at break is achieved when the laser fluence is around 8 J/cm^2^, while the maximum increase in tensile strength occurs at approximately 10 J/cm^2^. Using femtosecond laser surface irradiation, this study compares the maximum enhancement in the tensile strength of titanium alloy base materials, which is approximately 8.54%, and the maximum increase in elongation at break, which reaches 25.61%. In addition, the results verify that cracks in tensile fractures of TC4 start from the middle, while laser-induced fracture cracks occur from both ends.

## 1. Introduction

The mechanical properties of metallic materials, which are vital components extensively employed in engineering applications, play a pivotal role in ensuring the safety and reliability of numerous applications. In recent years, femtosecond laser technology has gained widespread attention in the field of surface modification of metallic materials [1,2,3,4,5]. Through the manipulation of femtosecond laser process parameters, precise control of micro-nanostructures on metal surfaces can be achieved, thereby resulting in noteworthy enhancements in their mechanical properties.

TC4 is renowned for its exceptional characteristics, such as high strength, light weight, corrosion resistance, high-temperature resistance, and biocompatibility, and has extensive applications in the aerospace, medical, chemical, automotive, and sports equipment fields [6,7]. With research on TC4 deepening in recent years, researchers have systematically studied its mechanical properties and fracture behavior. The Johnson–Cook failure model, widely applied in the field of metal impact fractures, was utilized by Zeling Zhao [8] to investigate the deformation mechanism of TC4 alloys; through a finite element simulation of the specimen, the entire process from crack initiation to fracture was obtained. A crack in a smooth specimen is first generated in the center. As plastic strain accumulates, cracks propagate on both sides until the specimen fractures. In notched specimens, cracks originate from both sides and propagate towards the center until fracture occurs. Wang [9] assessed the influence of notch depth on a low-stress brittle fracture of a TC4 substrate, revealing an escalating trend of crack formation with increasing notch depth. This occurrence is influenced by stress concentration and the combined impact of stress field intensity.

Forging, heat treatment, welding, and other heat treatment methods are widely employed in the process of enhancing the mechanical properties of metals. However, these traditional processing methods may not be suitable for certain use cases, such as with components that are challenging to disassemble. Therefore, non-heat treatment processing methods have become an indispensable choice in such scenarios. Research interest in laser-induced material structure modification or the improvement of various properties has increased in recent years amongst scholars worldwide. J. Bonse [10,11,12,13,14,15] studied the formation of femtosecond laser-induced periodic surface structures (LIPSSs) on different materials in an air environment, which revealed specific LIPSS patterns based on the laser polarization direction. Jin et al. [16] explored the fatigue fracture morphology of TC4 samples at the three damage stages of crack initiation, crack propagation, and instantaneous fracture by using laser-induced TC4 surfaces. The experiments showed that the main reason for the improvement in the strength of TC4 surface modification was the reduction in the grain size of TC4 using laser irradiation. Luo et al. [17] identified grain dislocation in TC4 by combining laser shock strengthening with shot peening. They discovered that TC4 grain dislocation and a dislocation-gradient microstructure layer could extend crack initiation and propagation lives by 184% and 45%, respectively. Li et al. [18] refined the surface grains of TC4 with laser surface remelting, which can strengthen the surface of TC4 with complex shapes and increase the yield strength and tensile strength by 28% and 35%, respectively. The improvement of these properties is attributed to the formation of grain dislocation during the tensile deformation of surface grain refinement, and the dislocation of grain is accumulated on the heterogeneous interface, which yields strength improvement. Yao et al. [19] investigated the impact of CO_2_ laser-induced surface remelting on TC4. The results indicated rapid remelting and solidification after laser irradiation, resulting in varying degrees of improvement in the microhardness and elastic modulus in the remelted zone.

The extremely short interaction time with the material is an essential characteristic of laser processing, especially femtosecond laser processing. Femtosecond lasers significantly reduce the formation of a heat-affected zone due to their short thermal diffusion distance formed within the material. Researchers are dedicated to enhancing the corrosion resistance [20,21] and biocompatibility [22,23,24] of TC4 through laser surface treatment. Studies on the tensile properties of TC4 after femtosecond laser processing are limited.

In this study, a 500 μm TC4 substrate was chosen. By adjusting the femtosecond laser parameters during femtosecond laser irradiation, stress–strain curves under different irradiation conditions were obtained through tensile testing, enabling the analysis of their impact on tensile strength. The stress–strain curves obtained during tensile testing revealed that the coordinates corresponding to the highest tensile strength and maximum elongation at break were achieved at laser fluences (Fps) of 8 and 10 J/cm^2^, respectively. Specifically, post-femtosecond laser surface irradiation (LSI) resulted in a maximum increase in tensile strength of approximately 8.54% and a maximum enhancement in elongation at break of 25.61%, compared with original TC4. The yield strength showed no significant variation within the tested laser process range. In addition, we verify that cracks in tensile fractures of TC4 start from the middle, while laser-induced fracture cracks occur from both ends. Microscopic morphology and structural analysis of the fracture surfaces under different processing conditions were conducted, and the impacts on tensile performance were discussed based on laser processing principles.

## 2. Materials and Methods

### 2.1. Laser Processing and Sample Preparation

This study employed an ultrafast pulse laser system from Shenzhen Han’s Laser Company (Shenzhen, China) for the double-sided microprocessing of the central section of tensile standard specimens made from original TC4 substrate, as illustrated in Figure 1a. The laser system featured pulse characteristics with a wavelength of 1040 nm, a repetition frequency of 10 kHz, and a pulse duration of 160 fs. The beam diameter was 28 μm, and an F-Theta lens with a focal length of 50 mm was utilized. A TC4 standard substrate with a length (L_0_) of 21 mm, a width (W) of 7 mm, and a thickness (T) of 500 μm, as shown in Figure 1b, was used with a sample length of 70 mm.

Prior to experimentation, the surfaces of original TC4 substrate underwent fine polishing using 1200-grit sandpaper to eliminate minor protrusions; the original composition of TC4 is shown in Table 1. Subsequently, a polishing compound was applied for a smoother surface. The samples were then ultrasonically cleaned in anhydrous ethanol for 10 min before processing, and the TC4 surfaces were rinsed with water to remove any surface impurities.

During the femtosecond laser processing, a linear scanning method was employed to irradiate the TC4 processing area, with a laser scanning interval of 10 μm. The average Fp in the laser spot was calculated using Formula (1):(1)$$Fp=\frac{4P}{f\pi D^{2}}$$

In Formula (1), *P* represents the laser power, *f* is the laser wavelength, and *D* denotes the beam diameter. The laser power during processing was deduced using this formula. In addition, variations in scanning speed were implemented to assess changes in the mechanical properties of TC4 under different laser processing parameters.

### 2.2. Tensile Testing

In this paper, a Zwick 8404 L (Ulm, Germany) tensile testing machine was selected to complete the tensile fracture test of TC4, as shown in Figure 2a. This testing machine boasts a maximum tensile force of 50 kN, with strain gauges employed throughout the entire tensile process. Uniaxial tensile tests were conducted on TC4 standard specimens at a testing speed of 0.1 mm/min to characterize their tensile performance. Figure 2b presents SEM observations of the tensile fracture interface of Figure 2a.

There were variations in the tensile properties of thin TC4 substrate after irradiation under different laser processing parameters. For this reason, adjustments were made to the laser fluence and scanning speed applied to the TC4 standard specimens. Each set of tensile specimens underwent three tests, and the average of the three tests was used. Significant changes in tensile strength and elongation at break of the thin TC4 substrate were observed after LSI. Engineering strain and engineering stress during the tensile process were calculated using Formula (2):(2)$$\left\{\begin{array}{c} \varepsilon_{eng}=\frac{∆L}{L}\\ \sigma_{eng}=\frac{F}{wT} \end{array}\right.$$

In Formula (2), *F* represents the tensile force, Δ*L* is the tensile displacement, *L* is a gauge length of 21.0 mm, *w* is a gauge width of 7.0 mm, and *T* is the material thickness. Elongation is calculated as *ε_eng_* × 100%.

### 2.3. Surface Morphology Characterization and Elemental Analysis

The morphology of laser-irradiated TC4 surfaces was characterized using a XploreCompact EM-30 (Oxford, UK) Plus desktop scanning electron microscope from COXEM. Elemental composition analysis was conducted using the accompanying energy-dispersive X-ray spectrometer. Moreover, the formation of diffraction peaks on thin TC4 substrates after laser irradiation was observed using a smartlab X-ray diffractometer manufactured by Rigaku (The Woodlands, TX, USA).

## 3. Results

### 3.1. Analysis of Tensile Fracture Morphology

The mechanism of femtosecond laser-induced surface modification in metals primarily involves non-thermal processes, electron excitation, non-thermal evaporation, self-focusing effects, and microscopic control of crystal structures and grain boundaries. In this study, the manipulation of femtosecond lasers was employed to achieve double-sided processing on thin TC4 by adjusting the processing parameters, thereby achieving control over the tensile properties [25] of TC4.

In this study, the laser scanning speed was set to 1000 mm/s, with the Fp varying from 2 to 16 J/cm^2^ in intervals of 2 J/cm^2^. Furthermore, the Fp was set to 10 J/cm^2^, with the laser scanning speed ranging from 600 to 1400 mm/s in intervals of 200 mm/s. Figure 3 illustrates the uniaxial tensile comparison tests between original TC4 and LSI under varying Fp conditions. The laser scanning power during femtosecond laser processing was calculated based on Formula (1) for laser fluence. Figure 3a depicts the stress–strain curves under the LSI process, with the Fp ranging from 2 to 10 J/cm^2^. Figure 3a shows that, during this phase, the tensile strength of the TC4 gradually increased with the rise in Fp. Moreover, the elongation at fracture in the TC4 after LSI increased compared with that of original TC4. Notably, the elongation at fracture was maximized when the Fp was 8 J/cm^2^, which showed a 25.61% improvement over original TC4. Figure 3b illustrates the stress–strain curves as Fp increased from 8 to 16 J/cm^2^ during the LSI process. The tensile strength exhibited an increasing trend followed by a decrease with the variation in Fp. The maximum tensile strength was achieved with an Fp of 10 J/cm^2^, which represented an 8.54% improvement compared to that of original TC4. As the Fp continued to increase, the tensile strength gradually decreased. With an Fp of 14 J/cm^2^, the tensile strength of the laser-treated TC4 was comparable to that of original TC4. When Fp exceeded 16 J/cm^2^, the tensile strength of the laser-treated TC4 was slightly lower than that of original TC4, and the elongation at fracture also significantly decreased.

Figure 4 presents the uniaxial tensile comparison tests between original TC4 and LSI under changing scanning speeds. Figure 4a,b, respectively, illustrate the stress–strain curves for scanning speeds ranging from 600 to 1000 mm/s and from 1000 to 1400 mm/s during LSI and for original TC4. In Figure 4a,b, the impact of laser scanning speed variation on the tensile strength is not apparent. However, at a scanning speed of 1400 mm/s, the elongation at fracture of the laser-treated TC4 was comparable to that of original TC4. Within the selected range of laser scanning speeds, a noticeable increase in elongation at fracture was observed as the scanning speed decreased.

Figure 3 and Figure 4 illustrate the surface modification of a 500 μm thin TC4 induced by femtosecond laser processing. After uniaxial tensile testing, it was observed that the process resulted in enhanced strength and ductility. The primary reason for this improvement lies in the rapid temperature changes that occurred on the metal surface during laser treatment, which may have introduced residual stresses. By optimizing the laser processing parameters, residual stresses can be minimized. In addition, laser treatment may induce localized melting and subsequent solidification on the metal surface, facilitating the formation of alloyed regions or altering the surface’s chemical composition. Through control of the laser processing parameters, it is possible to adjust the metal surface’s chemical composition, thereby improving its tensile properties.

### 3.2. Microscopic Analysis of Tensile Fracture

In the tensile tests, fractures typically occurred in regions with weaker microstructures, and the fractures can effectively reveal fracture mechanisms, modes, and causes in alloys. The fracture morphology of the tensile specimens was measured and analyzed to better investigate the impact of femtosecond laser irradiation on the mechanical properties of TC4 [8].

Scanning electron microscopy was conducted to capture images of the thin TC4 tensile specimens before and after femtosecond laser irradiation, in order to observe the surface morphology near the fracture section, as shown in Figure 5. The fractography reveals that the tensile fractures of TC4 exhibit a dimple fracture [26,27].

Figure 5 illustrates the tensile fracture morphologies of original TC4, as well as specimens with an Fp of 6 and 10 J/cm^2^. All thin TC4s exhibited various-sized dimples on the surface after fracture, which resulted from the generation and merging of micropores. In addition, the upper part of the surfaces after laser irradiation was visibly altered, which implies the formation of microstructures on the thin TC4 surface due to the energy distribution of the laser spot. Moreover, the surface roughness noticeably increased with the rise in laser fluence. Compared with those in original TC4, the dimples on the fracture surface after LSI are deeper, and their sizes undergo significant changes.

In order to better evaluate the effect of laser parameters on the tensile fracture of TC4, the number and size of the dimples observed under SEM were analyzed using an open-source image processing software [28] (ImageJ2 Fiji). Based on Figure 5, the dimples of TC4 before and after laser treatment were measured and analyzed using ImageJ software, and the histograms shown in Figure 6 were drawn. The “D” in Figure 6 represents the average size of the dimple, and the extreme values of the dimple are denoted by “S_max_” and “S_min_”, respectively. Using a scanning speed of 1000 mm/s and Fps of 6 and 10 J/cm^2^, the number and size of dimples were significantly different from those of the original TC4. Specifically, 170 dimples were observed, as shown in Figure 6a, with an average size of 3.990 ± 2.291 μm, and some of them had less distinct dimples with clearer boundaries and elliptical shapes, as shown in Figure 6b,c. In Figure 6b, 112 dimples were observed, with an average size of 5.434 ± 2.707 μm, while in Figure 6c, only 102 dimples were observed. The average size of the dimples shown in Figure 6c was different from that in the previous two figures and also showed an increasing trend. The extreme value of dimples followed a similar change trend to the average size.

Figure 7 shows scanning electron microscope images of the tensile fracture surfaces of the TC4 base and the tensile fracture surfaces under varying laser scanning speeds. As shown in Figure 7b,c, the dimples generated by tensile fractures were deeper and denser than those in Figure 6a, which indicates that femtosecond laser irradiation on thin TC4 substrates can influence tensile fracture behavior. Figure 7d,e show local magnifications at scanning speeds of 600 and 1400 mm/s, respectively. Comparing the clear magnifications in Figure 7d,e shows that the dimples were significantly larger and deeper at a scanning speed of 600 mm/s. Large and deep dimples are conducive to improving the ductility of metal materials. Combining the stress–strain curves obtained from tensile tests on thin TC4 substrates under different laser parameters, as shown in Figure 3, shows that the evaluation indicators of tensile performance exhibit different changing trends with variations in the laser fluence and scanning speed.

Tensile strength and elongation at break showed noticeable changes with increased energy density, and their changing trends were similar. Both increased initially and then decreased with the rise in laser fluence, with the peak tensile strength occurring around a laser fluence of 10 J/cm^2^, and the peak elongation at break around a laser fluence of 8 J/cm^2^. In addition, when the energy density was too high, the tensile strength became lower than that of the original TC4. Moreover, Figure 4 demonstrates that reducing the scanning speed can increase the elongation at break without significantly affecting the tensile strength.

Figure 8 depicts a measurement analysis histogram of the dimple in Figure 7, obtained using ImageJ image processing software. The 58 dimples in Figure 8a had an average size of 3.213 ± 2.143 μm. The number of dimples in Figure 8b is slightly lower than that in Figure 8a, with an Fp of 10 J/cm^2^ and scanning speed of 600 mm/s. Compared with the dimples Figure 8a, the average dimple size increased significantly. In Figure 8c, the numbers and average size of dimples are different from the previous observations: with an Fp of 10 J/cm^2^ and scanning speeds of 1400 mm/s, 70 dimples were observed, which was much more than in the original TC4 under the same observation conditions, and the average size of dimples decreased significantly.

Figure 9 shows the location of the fracture surface of thin TC4 and LSI-treated TC4 after tensile fracture. As Figure 9a shows, cracks in thin TC4 are initiated from the middle and extend to a position approximately 45 μm away from the surface. As shown in Figure 9b, after LSI treatment, cracks in TC4 initiated from the processed surface, and the crack length could exceed 75 μm. This is because the crystal interior of the TC4 substrate contains microdefects and dislocations, serving as initiation points for cracks [29]. LSI alters the crystal structure of TC4 and the grain boundary characteristics. The grain size of TC4 may decrease, and the grain boundaries become denser. Moreover, LSI may change residual stresses in TC4 by adjusting the parameters appropriately, which aids in controlling the path of crack formation. This surface modification helps enhance the material’s surface strength and toughness, making crack initiation from the surface easier during tensile fracture. Furthermore, the high energy density induced by femtosecond lasers may alter the grain orientation and boundary structure, which leads to grain refinement on the TC4 surface, contributing to finer grains on the surface and coarser grains internally. This grain structure densifies and strengthens the grain boundaries, altering the movement of dislocations [18,30]. Dislocations accumulate at heterogeneous interfaces, contributing to an overall improvement in the tensile strength of the material. This strength improvement mainly occurs from the yield stage to the fracture stage of the material, while the impact on the elastic stage is relatively small, which is consistent with the conclusions in Figure 3 and Figure 4.

Figure 10 presents the energy dispersive spectroscopy analysis results for the fractured surface, on the surface and internally, under LSI with an Fp of 10 J/cm². In Figure 10a, which corresponds to the laser-treated surface, and Figure 10b, which represents the internal section of the thin TC4, the carbon, aluminum, and vanadium elements exhibit similar concentrations on the surface and internally. However, a noticeable difference in the oxygen element was observed. Figure 10a presents an enriched oxygen content on the surface after LSI treatment, whereas Figure 10b indicates an absence of oxygen. This suggests that LSI treatment forms a layer of titanium oxide (TiO_x_) on the surface of the thin TC4. This phenomenon was attributed to femtosecond laser ablation altering the microscopic morphology of the thin TC4. In addition, during the femtosecond laser ablation of TC4 conducted in an atmospheric environment, the surface of the thin TC4 experienced varying degrees of mild oxidation.

### 3.3. X-ray Diffraction Analysis

Figure 11 illustrates the X-ray diffraction (XRD) spectra analysis of the TC4 substrate using various Fps (1000 mm/s) under LSI. XRD analysis was conducted in the range of 20° to 80° in this study. Figure 11a shows that all diffraction peaks appearing on the TC4 substrate in the range of 30° to 80° systematically match with the XRD curves after LSI treatment. Furthermore, new structural peaks were generated on the material surface post-LSI treatment. Analysis of the positions of the diffraction peaks indicated that the oxide formed on the TC4 surface after LSI treatment was TiO_2_. Apart from the TC4 substrate, structural peaks with varying TiO_2_ peak intensities appeared between 39.5° and 63°~64°. The peak intensity was highest at an Fp of 16 J/cm^2^, which implies that an increase in Fp promotes the generation of TiO_2_. Detailed plots at specific diffraction angles were set, as shown in Figure 11b: with θ in the range of 38~39° and a scanning speed of 1000 mm/s, compared to the original TC4 with 6 and 10 J/cm^2^ samples, a deviation in the diffraction peak was observed. Figure 11b also reveals a significant phenomenon: After LSI treatment, the peak broadening of 6 and 10 J/cm^2^ samples increased significantly, the peak broadening of 10 J/cm^2^ was a little higher than 6 J/cm^2^. Peak broadening is a measure of the residual elastic energy stored in the strain field of dislocations [31,32], which is the source of residual stresses; these results are consistent with previous studies and verify that dislocation formation is exacerbated during rapid cooling after laser surface treatment [33]. Therefore, laser treatment can generate residual stresses and exacerbate the formation of dislocations. The exacerbation of dislocation contributes to the improvement of the strength of TC4. In this study, 10 J/cm^2^ was identified as the threshold for changes in tensile strength, and when the Fp went beyond 10 J/cm^2^, the tensile strength, surprisingly, decreased.

## 4. Conclusions

In this study, by adjusting laser processing parameters, the tensile properties of TC4 were improved by femtosecond laser irradiation on surfaces treated in the air. The results indicated that the following:

Cracks in thin TC4 initiated from the middle and extended to a position approximately 45 μm away from the surface; after LSI treatment, cracks in TC4 initiated from the processed surface, and crack lengths could exceed 75 μm. The main reason was the high energy density induced by the femtosecond laser altering the grain orientation and boundary structure, which led to grain refinement on the TC4 surface. Increasing Fp induced significant changes in both the tensile strength and elongation at break of TC4 within the explored range of laser processing parameters. At a scanning speed near 1000 mm/s and an Fp of 8 J/cm^2^, the elongation at break of TC4 reached its maximum enhancement, which showed a 25.61% improvement compared with that of the substrate. Similarly, at a scanning speed of 1000 mm/s and an Fp of 10 J/cm^2^, the tensile strength experienced the greatest improvement, resulting in an 8.54% increase compared to that of the substrate. However, beyond 10 J/cm^2^, both the tensile strength and elongation at break of the TC4 after LSI started to significantly decline.

The spot energy of femtosecond lasers on the surface of thin TC4 formed microstructural topography; with the increase in laser fluence, the surface roughness significantly increased. Moreover, compared to the TC4 substrate, the dimples on the tensile fracture surface after LSI were deeper, and the size of the dimples also underwent noticeable changes. The high energy density induced by lasers alters the orientation of grains and the structure of grain boundaries, resulting in grain refinement on the TC4 surface, forming a structure with finer grains on the surface and coarser grains internally. This grain structure densifies and strengthens the grain boundaries, altering the movement of dislocations, with dislocations accumulating at heterogeneous interfaces, thereby improving the overall tensile strength of the material.

In an atmospheric environment, femtosecond laser ablation of the surface of thin TC4 led to varying degrees of mild oxidation. With Fp promotion, the generation of TiO_2_ improved. Laser treatment can generate residual stresses and exacerbate the formation of dislocations. The exacerbation of dislocation contributed to the improvement of the strength of TC4. In this study, 10 J/cm^2^ was identified as the threshold for changes in tensile strength; when exceeded, the tensile properties of TC4 are reduced.

## Figures and Tables

**Figure 1 micromachines-15-00152-f001:**
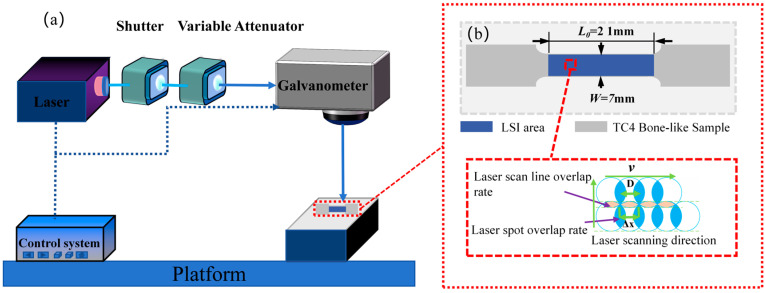
Femtosecond laser system for processing TC4. (**a**) Femtosecond laser system layout. (**b**) Femtosecond laser irradiation of TC4 surface.

**Figure 2 micromachines-15-00152-f002:**
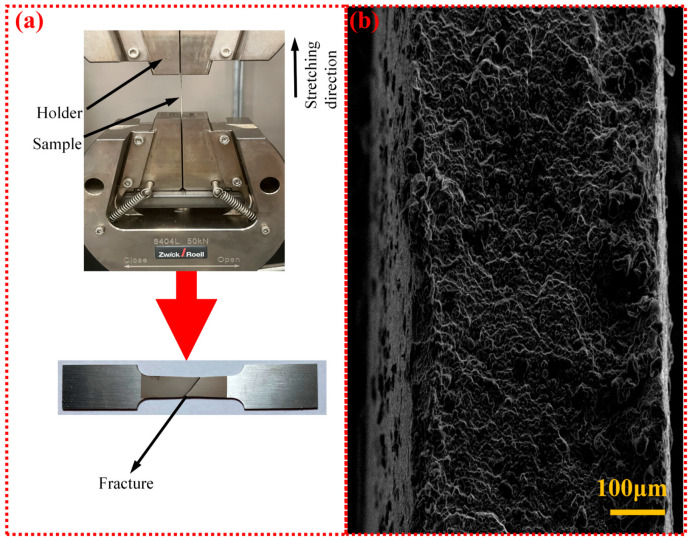
Uniaxial tensile test of thin TC4: (**a**) detail of tensile fracture process and (**b**) SEM image of tensile fracture section.

**Figure 3 micromachines-15-00152-f003:**
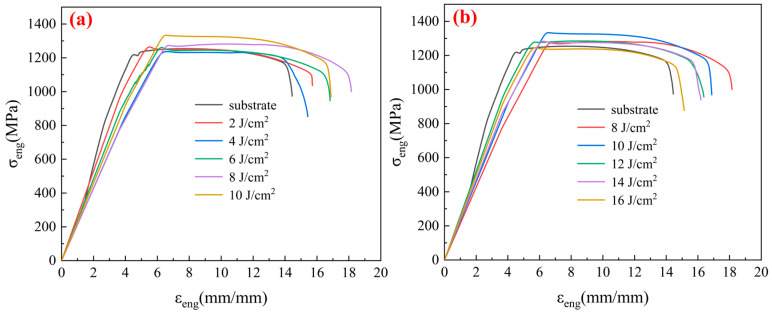
Uniaxial tensile performance of the TC4 sample with a laser fluence of (**a**) 2–10 J/cm^2^ and (**b**) 8–16 J/cm^2^.

**Figure 4 micromachines-15-00152-f004:**
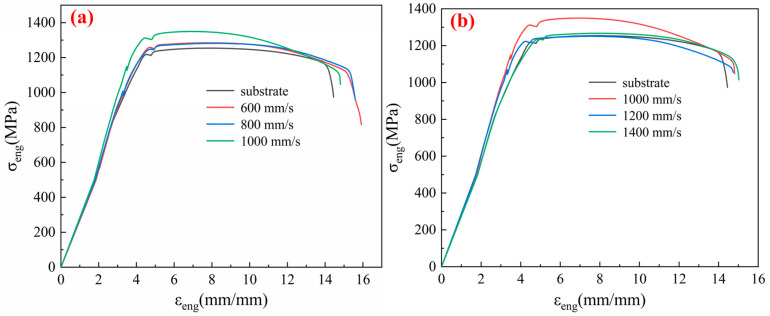
Uniaxial tensile performance of the TC4 sample at an Fp of 10 J/cm^2^, with a scanning speed variation of (**a**) 600~1000 mm/s compared with TC4 substrate and (**b**) 1000~1400 mm/s compared with TC4 substrate.

**Figure 5 micromachines-15-00152-f005:**
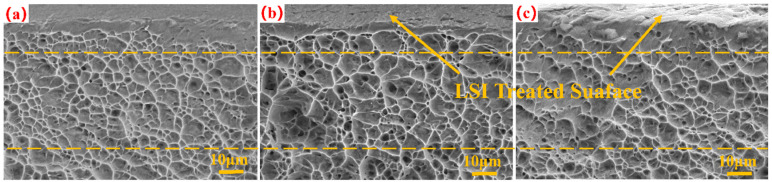
Using a scanning speed of 1000 mm/s, SEM images of original sample and laser-treated samples. (**a**) Original TC4 substrate; (**b**) Fp of 6 J/cm^2^; and (**c**) Fp of 10 J/cm^2^.

**Figure 6 micromachines-15-00152-f006:**
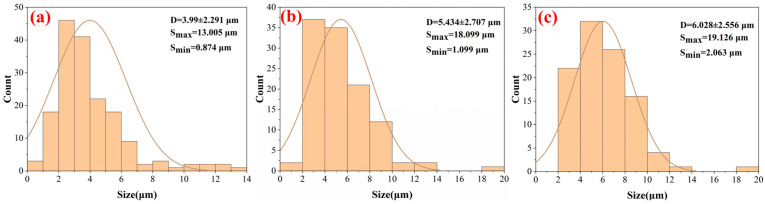
Using a scanning speed of 1000 mm/s, histograms drawn by ImageJ of original TC4 and laser-treated samples. (**a**) Original TC4 substrate, (**b**) Fp of 6 J/cm^2^, and (**c**) Fp of 10 J/cm^2^.

**Figure 7 micromachines-15-00152-f007:**
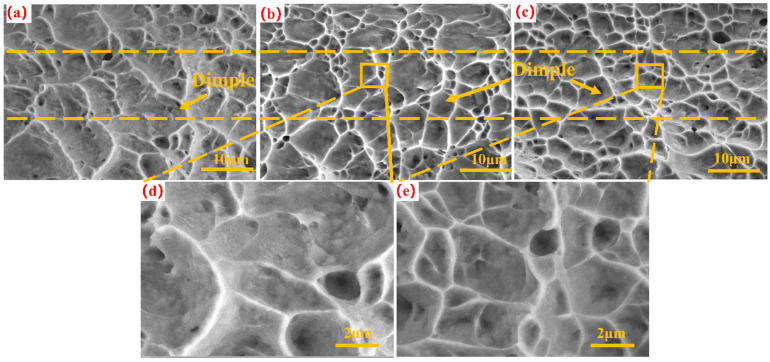
Using an Fp of 10 J/cm^2^, SEM images of original sample and laser-treated samples with scanning speed variations. (**a**) Original TC4 substrate, (**b**) 600 mm/s scanning speed, and (**c**) 1400 mm/s scanning speed; (**d**) local magnification of (**b**,**e**) local magnification of (**c**).

**Figure 8 micromachines-15-00152-f008:**
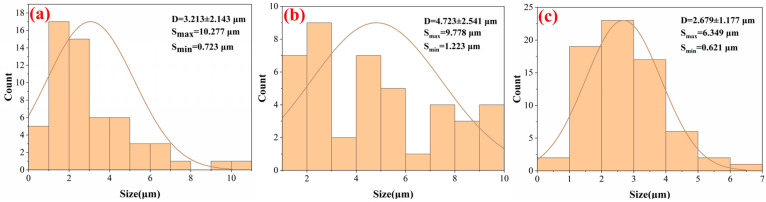
Using an Fp of 10 J/cm^2^, histograms drawn by ImageJ of original TC4 and laser-treated samples with scanning speed variations. (**a**) Original TC4 substrate, (**b**) 600 mm/s scanning speed, and (**c**) 1400 mm/s scanning speed.

**Figure 9 micromachines-15-00152-f009:**
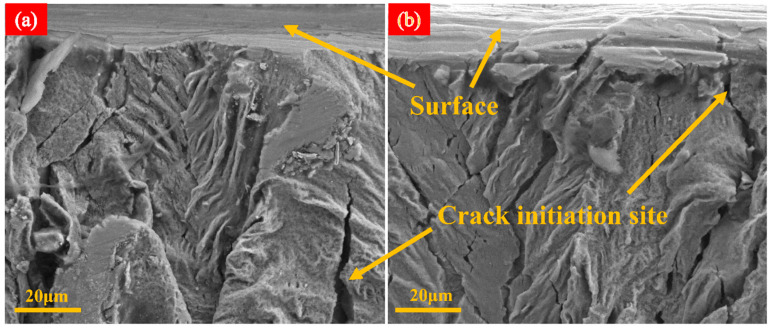
The location of the fracture surface of original TC4 and laser-treated TC4 after tensile fracture: (**a**) original TC4 substrate; (**b**) after laser treatment.

**Figure 10 micromachines-15-00152-f010:**
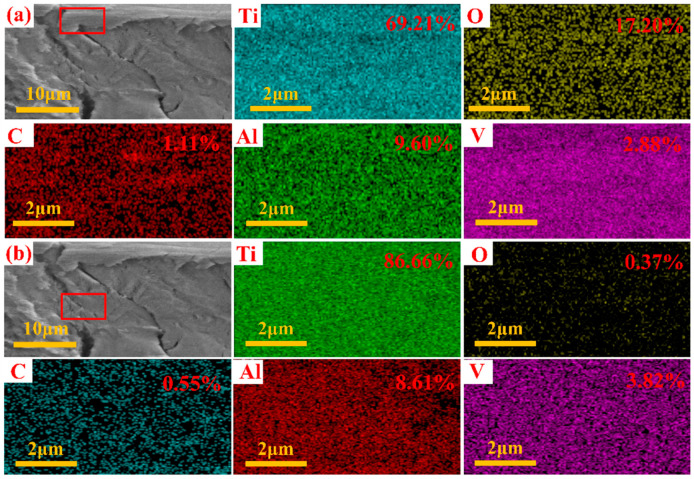
Using an Fp of 10 J/cm^2^, EDS images of surface fracture and interior of fracture surface; the upper right corner is the amount of the element (**a**) on the laser-treated surface, (**b**) in the interior.

**Figure 11 micromachines-15-00152-f011:**
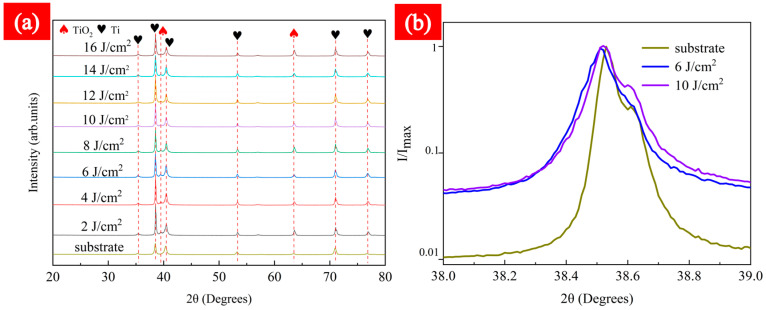
Using a scanning speed of 1000 mm/s, with Fps varying from 2 to 16 J/cm^2^, XRD comparison with TC4 substrate. (**a**) θ = 20~80° and (**b**) X-ray peaks for TC4 substrate, including a 6 J/cm^2^ sample and a 10 J/cm^2^ sample, with a θ of 38~39°.

**Table 1 micromachines-15-00152-t001:** Chemical composition of original TC4 (mass fraction).

Element	Ti	Fe	N	H	O	Al	C	V
Quality Proportion (%)	margin	0.3	0.05	0.015	0.02	5.5–6.8	0.1	3.5–4.5

## Data Availability

Data are contained within the article.

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
