# Peer review of "Influence of Femtosecond Laser Surface Modification on Tensile Properties of Titanium Alloy"

_micromachines, 2024, doi:10.3390/mi15010152_

Round 1

Reviewer 1 Report

Comments and Suggestions for Authors

Influence of Femtosecond Laser Surface Modification on the Tensile Properties of TA6V

This manuscript communicates some interesting aspect of surface engineering for enhancing surface tensile properties using ultrafast laser pulses. It is fairly well written with reasonable clarity and technical soundness. This work presents surface irradiation of a Ti-alloy. Then, authors perform a creep/tensile test that exhibits some interest and might be useful for future works. I reckon the content of this manuscript may have the potential to benefit the scientific and engineering communities in the relevant fields.

I do have a strong feeling that in terms of novelty and impact the manuscript in its current is not up to par. The authors may want to argue about this, do so, and make the arguments appear in the manuscript. In my opinion, the manuscript lacks studies and discussions. Thus, I cannot recommend it for publication until a major revision is performed.

Major issues:

·         Introduction session, paragraph 3, is this relevant to this paper? If the answer is affirmative, please state how? I fail to see the connections between what the author reviews in this paragraph with what the author reports in this paper.

·         References are needed for the last part of the Introduction session, and across the entire paper. There is not a single reference whatsoever after the Introduction session. This cannot be, for a referred, peer reviewed article. Has the author pursued a literature survey before launching his/her own research? If not yet, I recommend following articles to start with:

DOI: 10.2351/1.4967013

DOI: 10.1038/s41598-023-40283-6

DOI: 10.1063/5.0052510

More relevant articles can be found through the citation record of these publications.

·         In section Mat & Met, the femtosecond laser specifications should be given, model and manufacturer

·         In the Results section, the biggest thing is Figure 4 is a duplicate of Figure 3. Therefore, without having access to the data, the review has been quite difficult to process.

·         In section 3.1 the author reports an improvement of tensile fracture. This is fine I believe there’s a global effect of the laser irradiation, even the laser impact depth is quite limited, I assume. In 3.2, the results reported around Figures 5 and 6 seem to suggest an influence of femtosecond laser irradiation can be found deep in the bulk (50 – 100 µm deep as shown in figure 5, for instance). I am not convinced that microstructural impact of femtosecond laser irradiation can be found in such a region, far low under the surface. I would suggest the author to further investigate in details. First question to answer is where the fractures were initiated, from laser impacted side or from the other side? Laser irradiation prevents the fractures to start at the beginning of the tensile test, or prevents the sample to break at the end of the test?

·         In 3.2 and 3.3 the author reports TiOx formation – I agree this is evident from EDX and XRD. Notwithstanding, I am not sure the formation of TiOx is relevant to tensile stress. The author has to elaborate on this.

Suggestion: if the author has access to XRD, attempts can be made to deduce dislocation densities before/after laser irradiation, i.e. see DOI: 10.1063/5.0052510

Minor issues:

Consistency of the terminology “Ti-6Al-4V” or “TC4”?

Format issues in the Reference section. Some of the entries are not properly displayed (ref 19, 24, 25, for example).

Comments on the Quality of English Language

English issue, some typos, and incomplete sentences are found in the manuscripts

Reviewer 2 Report

Comments and Suggestions for Authors

In this manuscript, the authors conducted experimental investigations on improving the tensile properties of Ti-6Al-4V by femtosecond laser processing. The obtained stress-strain curves demonstrated that femtosecond laser processing applies a positive effect on enhancing tensile properties at the moderate energy level. The SEM/EDX/XRD characterization show the presence of the oxidation process during laser processing and its effect on overall mechanical properties. The obtained results are interesting to the broad readership of Micromachines. However, there is a lot of key information and discussion missing. To make this manuscript acceptable, the authors are required to address the following comments well.

1. The quality of English writing should be improved. There are lots of grammar issues throughout the manuscript.

2. The introduction part is not sufficient. The literature review on applications of laser to improve the mechanical performance of metal/alloy materials should be included.

3. In the introduction part, the authors mentioned that yield strength showed no significant variation. However, there is no discussion provided in the manuscript on it. The stress-strain curves provided are not clear enough to tell the variation of yield strength.

4. Please double-check the manuscript and make sure to add SPACE between value and unit.

5. There is no Figure.2(b) provided in the manuscript. But it is cited in line 127.

6. The plots in Figure 4 are identical to those in Figure 3. Please add the correct plots.

7. The discussion on the mechanism that femtosecond laser processing can improve tensile performance is not sufficient. What is the mechanism?

8. The oxidation process is more likely to occur during laser irradiation rather than post-exposure to the ambient. (line 240-244)

9. (Line 260-265) The authors claimed that a certain concentration of TiO2 is beneficial for improving tensile strength. But lower or higher concentration will undermine that behavior. What is the mechanism for this explanation? TiO2 is more plastic than Ti-6Al-4V?

Based on the abovementioned comments, this manuscript is recommended for major revision. A revised manuscript is required. 

Comments on the Quality of English Language

Please check the comments in the Comments and Suggestions section. 

Round 2

Reviewer 1 Report

Comments and Suggestions for Authors

Review report (round 2) for Influence of Femtosecond Laser Surface Modification on the Tensile Properties of Ti-6Al-4V

The quality of the manuscript after the 1st revision is better than the original version. It is clearer and more convincing, I reckon.  There are still quite a few major weaknesses that have to be improved before the manuscript can be considered for publication.

1.       The title should be changed to TC4 if the authors opts to TC4 for the text.

2.       The chemical composition of TC4 should be given.

3.       I advise an inset of optic macroscope and/or microscope image, or alternatively a SEM micrograph at low Mag., to be added in Figure 2 to highlight the entire fracture section. A global view of the initiations of the cracks would be highly appreciated. The author tries to highlight the crack initiation of original TC4 and laser treated TC4 in Figure 7 – the micrographs in Figure 7 is too localized and hard to visualize whether they are representative and with statistic importance.

4.       For Figure 5 and Figure 6, the differences are not evident. I would suggest some sort of image analysis such as dimple count with ImageJ to be performed. A histogram of dimple distribution can enhance the author’s arguments. Check out this for example: https://doi.org/10.1080/22020586.2019.12073197

And also, from following publications:

- Saltykov, S. A. Stereometric Metallography. Metallurgizdat, 1970.

- Cuzzi, J. N.; Olson, D. M. Meteoritics & Planetary Science 2017, 52, 532.

- Lopez-Sanchez, M.; Llana-Funez, S. Journal of Structural Geology 2016, 93, 149.

5.       I think I suggested in the first review report (I don’t have access to this report during this holiday season as it is locked in my office), the dislocation density can be evaluated by X-ray diffraction analysis. Peak broadening is a measure of residual elastic energy stored in the strain field of dislocations. As example can be found in https://doi.org/10.1063/5.0052510. As the authors has means of XRD, one could try to quantify the dislocation density modification. This could be helpful for explaining how come 10J/cm2 is producing the best peening effect and how come there is an optimal scan speed.

6.       More than half of the references are from Asian academy and mostly from China. It a good sign that there are lots of relevant activities in this domain. I believe nonetheless there are similar research going on in the rest of the world. Given the fact MDPI is an international publisher and targets audience from the globe, I would suggest to add more international entries

Comments on the Quality of English Language

minor issues such as line 40: no capital letter after comma;

line 88: TC4 original TC4 

Reviewer 2 Report

Comments and Suggestions for Authors

The authors have answered all comments well. The current version is recommended for publication.
